# Alterations of the Sympathoadrenal Axis Related to the Development of Alzheimer’s Disease in the 3xTg Mouse Model

**DOI:** 10.3390/biology11040511

**Published:** 2022-03-26

**Authors:** Alicia Muñoz-Montero, Ricardo de Pascual, Anabel Sáez-Mas, Inés Colmena, Luis Gandía

**Affiliations:** Instituto Teófilo Hernando and Departamento de Farmacología y Terapéutica, Universidad Autónoma de Madrid, C/ Arzobispo Morcillo, 4, 28029 Madrid, Spain; alicia.munnozm@uam.es (A.M.-M.); ricardo.pascual@uam.es (R.d.P.); ansama96@gmail.com (A.S.-M.); ines.colmena@uam.es (I.C.)

**Keywords:** Alzheimer’s disease, 3xTg mouse model, sympathoadrenal axis, chromaffin cell, autocrine-paracrine modulation, voltage-dependent calcium channels

## Abstract

**Simple Summary:**

Alzheimer’s disease (AD), the most common form of dementia, is becoming a global health problem and public health priority. In the advanced stages of AD, besides the initial cognitive symptoms, behavioral problems, particularly agitation and aggressiveness, become prevalent in AD patients. These non-cognitive symptoms could be related to alterations in the regulatory mechanism of the sympathetic nervous system. In this study, we used chromaffin cells (CCs) isolated from the adrenal gland of 3xTg (an AD mouse model) mice to characterize potential alterations in the regulation of the responses to stress mediated by the secretion of catecholamines. We compared these regulatory mechanisms in mice at two different ages: in 2-month-old mice, where no AD symptoms were observed, and in mice over 12 months of age, when AD-related cognitive impairment related was fully established. We found that the modulation of neurotransmitter release was stronger in CCs isolated from the adrenal medulla of 3xTg mice older than 12 months of age, an effect likely related to disease progression as it was not observed in CCs from age-matched wild-type (WT) mice. This enhanced modulation leads to an increased catecholamine release in response to stressful situations, which may explain the non-cognitive behavioral problems found in AD patients.

**Abstract:**

Alzheimer’s disease (AD), the most common form of dementia, is becoming a global health problem and public health priority. In the advanced stages of AD, besides the initial cognitive symptoms, behavioral problems, particularly agitation and aggressiveness, become prevalent in AD patients. These non-cognitive symptoms could be related to a noradrenergic overactivation. In this study, we used chromaffin cells (CCs) isolated from the adrenal gland of 3xTg AD model mice to characterize potential alterations in the autocrine-paracrine modulation of voltage-dependent calcium channels (VDCCs), which in turn serve to regulate the release of catecholamines. We used mice at the presymptomatic stage (2 months) and mice over 12 months of age, when AD-related cognitive impairment was fully established. We found that the modulation of inward currents through VDCCs induced by extracellular ATP was stronger in CCs isolated from the adrenal medulla of 3xTg mice older than 12 months of age, an effect likely related to disease progression as it was not observed in CCs from age-matched WT mice. This enhanced modulation leads to increased catecholamine release in response to stressful situations, which may explain the non-cognitive behavioral problems found in AD patients.

## 1. Introduction

Alzheimer’s disease (AD) is the most common form of dementia; aging is the main variable linked to the development of this disease, which is becoming a global health problem and a public health priority, given that world’s elderly population is constantly growing.

The main pathogenic hallmarks of AD are the formation of senile plaques, due to the deposition of amyloid-beta (Aβ), and the formation of neurofibrillary tangles as a result of the hyperphosphorylation of the tau protein [1]. This results in synaptic deficits and synaptic loss, which seem to be correlated with the progression and severity of AD-associated memory loss and atrophy of the cerebral cortex [2]. The initial AD symptoms (i.e., cognitive impairment, memory loss, and learning deficits) appear to be related to the loss of cholinergic innervation in the cerebral cortex, leading to the formulation of the “Cholinergic Hypothesis of AD” [3,4] and the development of cholinesterase inhibitor-based therapies. However, as AD progresses, changes in the cholinergic, serotonergic, noradrenergic, dopaminergic, GABAergic, and somatostatinergic neurons are observed. Studies with transgenic mouse models of AD have revealed that although cholinergic neurotransmission appears to be the most vulnerable in the early stages of the disease, as AD progresses, glutamatergic terminals and finally the somewhat more resilient GABAergic terminals will be also affected [5]. Thus, the alteration of several neurotransmitter systems has been reported in advanced AD, which could be correlated with changes in the synthesis, storage or release of these neurotransmitters.

In patients suffering from AD, the progression of the disease and the subsequent increase in neurological limitations generates a stressful situation that makes them more vulnerable. During the last few years, different studies have suggested that the sympathoadrenal axis is also affected as AD progresses; therefore, there is a growing interest in studying the responses of this axis in these patients as well as in the transgenic models of AD. Peripheral sympathoadrenal axis activity is finely controlled at specific sites in the brain cortex and hypothalamus, and alterations in the release of exocytotic neurotransmitters may also occur at peripheral sympathetic neurons and adrenal medullary chromaffin cells (CCs). The CCs are integrated into the sympathoadrenal axis, which helps to maintain the body homoeostasis during both routine life and acute stress conditions [6]. These cells are widely used to explore neurotransmitter release and synaptic transmission, and are considered to be the amplifying arm of the sympathetic nervous system: the sudden and fast exocytotic release of their catecholamines—stored in noradrenergic and adrenergic cells—plays a fundamental role in the fight-or-flight response to stress [6]. More recently, CCs have also been used to determine the role of some neurodegenerative disease-related proteins in the regulation of cell excitability, ion channel currents, bulk exocytosis, and the fine kinetics of single-vesicle exocytotic events [7,8,9,10,11]. This has also been the case for some mouse models of neurodegenerative diseases (NDDs) [12,13,14,15] and autism [16]. Our studies in different mouse models carrying mutations linked to AD and other NDDs have provided results consistent with the hypothesis that the central alterations that occur in various NDDs are also manifested at the peripheral sympathoadrenal axis, impairing the fight-or-flight stress response in patients suffering from these diseases [17].

It should be noted that the physiological function of CCs in the sympathoadrenal axis is for the massive exocytotic release of the catecholamines epinephrine and norepinephrine into the circulation in response to stress [18]. To facilitate this massive and sudden release of catecholamines, CCs use an autocrine-paracrine mechanism to regulate exocytosis [19,20]. Catecholamines and other co-exocytosed vesicular components, such as ATP [21] or opioid peptides [22], modulate the activity of voltage-dependent calcium channels (VDCCs), i.e., a membrane delimited G protein-mediated inhibition of calcium channels that is relieved by voltage. Thus, for instance, when ATP binds to its purinergic receptor, the G protein couples to the channel, slowing down the current activation and decreasing its peak amplitude. The application of a strong positive depolarizing pre-pulse uncouples the G protein from the channel, and the current recovers its control profile [20,21], thus increasing Ca^2+^ influx to trigger massive catecholamine exocytosis.

In order to understand the pathogenesis of AD and, hence, search for new therapies, the mouse model of AD, 3xTg, has been widely used as a relevant experimental model. It has three mutations—APPSwe, PSEN1 with human M146V mutation, and the P301L mutation of tau—showing the typical hallmarks of the disease [23].

In the present study, we used CCs isolated from the adrenal glands of this mouse model to characterize possible alterations in the sympathoadrenal axis that could appear with the progression of AD. To this aim, we compared the modulatory effects exerted by extracellular ATP on inward currents through VDCCs in voltage-clamped CCs from wild-type (WT) and 3xTg mice at the presymptomatic (2 months) and symptomatic (>12 months) stages. In order to identify the possible mechanisms involved in the observed differences, the expression of the different VDCC subtypes and the levels of transcription of different purinergic receptors (P2Y and P2X) expressed in CCs were also characterized.

## 2. Materials and Methods

### 2.1. Animals

All procedures involving animals in this study were performed in accordance with the recommendations of the Ethics Committee from Universidad Autónoma de Madrid on the use of animals for laboratory experimentation, and following the ethical code and established guidelines from the European Union Directive (2010/63/EU) and Spanish legislation (RD 1201/2005 and 53/2013). Mice were housed under controlled conditions of temperature and light (12:12 h), and food and water were provided ad libitum.

Male 3xTg (129/C57BL6 background) mice were kindly donated by Dr. Javier García Sancho (Universidad de Valladolid, Spain), and contained the PS1M146V mutation (substitution of methionine and valine at codon 146), the “Swedish” mutation of the amyloid precursor protein (APPSwe), and the P301L tau mutation (substitution of proline by leucine at codon 301) under the transcriptional control of the Thy1.2 expression cassette [23]. The controls, male B6129SF2J mice (WT), were generously provided by Dr. Miguel Ángel del Pozo (CNIC, Madrid).

3xTg mice model develop cognitive impairment, starting at 6 months of age (Appendix A). Therefore, in this study, the mice were used at a presymptomatic stage (2 months, hereinafter referred to as 2 m) and at over 12 months of age (hereinafter referred to as 12 m), when the cognitive impairment related to AD was completely established.

All efforts were made to avoid animal suffering, and we used the minimum number of mice in search of significant differences between experimental groups. All work was performed under blinded conditions.

### 2.2. Genotype Determinations

Genomic DNA was extracted from mouse ear using the REDExtract-N-Amp Tissue Kit (Sigma-Aldrich, Madrid, Spain). Briefly, the tissue was incubated in extraction solution for 10 min at room temperature and then at 95 °C for 3 min. After this, neutralizing solution was added.

For polymerase chain reaction (PCR), the following specific primers were used: APP Fw 5′-GCTGCACCAGTTCTGGATGG-3′, Rv 5′-GAGGTATTCAGTCATGTGCT-3′; tau Fw 5′-GAGGTATTCAGTCATGTGCT-3′, Rv 5′-TTCAAAGTTCACCTGATAGT-3’; PS1 Fw 5′-AGGCAGGAGATCACGTGTTCAAGTAC-3, Rv 5′-CACAGGCACACTCTG ACATGCACAAGGC-3′. PCR products were analyzed by electrophoresis on a 1.8% agarose gel stained with REDTaq (Sigma-Aldrich, Madrid, Spain) to visualize the DNA bands.

### 2.3. Primary Cultures of Mouse Chromaffin Cells

Adrenal glands from WT and 3xTg mice were isolated and rapidly placed in ice-cold Locke solution ((in mM) 154 NaCl, 5.5 KCl, 3.6 NaHCO_3_, 10 Hepes, and 5.5 glucose (pH = 7.4)) after their sacrifice by cervical dislocation. The glands were fat trimmed, and the medullae of both glands were isolated from the capsule and the cortex and placed in a 1.5 mL tube containing 200 uL of Locke’s solution and papain (25 U/mL, Sigma-Aldrich) for tissue digestion at 37 °C for 26 min. This solution was exchanged with 1 mL of Dulbecco’s modified Eagle’s medium (DMEM; D6546, Sigma-Aldrich), repeating the exchange three times and finally leaving 120 μL of DMEM. Then, the medullae were mechanically digested by mincing them first with a 1 mL and then with a 200 μL micropipette tip. Finally, the residual medulla fragments were discarded and 10 μL drops of DMEM containing the cells were placed on poly-D-lysine-coated coverslips in 24-well plates. After a 1 h incubation (37 °C, water saturated, and 5% CO_2_ atmosphere), 500 μL of DMEM supplemented with 5% fetal bovine serum, 50 IU/mL penicillin, and 50 μg/mL streptomycin was added to each well. Experiments were conducted within the following 3 days.

### 2.4. RNA Extraction and Quantitative Reverse Transcription PCR (RT-qPCR)

The samples were homogenized by vortexing and pipetting with 250 µL of TRIzol reagent (Sigma-Aldrich). The homogenate was then mixed with 100 µL of chloroform and centrifuged for 15 min at 12,000 rpm in 4 °C conditions. The supernatant was recovered, mixed with 250 µL of isopropanol, and frozen overnight at −80 °C. The samples were centrifuged for 15 min at 12,000 rpm in 4 °C condition, and the supernatant was discarded. After complete supernatant evaporation, the pellet was resuspended with RNase free water and stored at −80 °C.

The total RNA extracted from the different tissues was quantified using a NanoDrop^TM^ spectrophotometer (ThermoFisher Scientific; Waltham, MA, USA), and 1 µg of each sample was converted into complementary DNA using a PrimeScript™ RT Reagent reverse transcription kit (Takara Bio Europe SAS; Saint-Gemain-en-Laye, France; Cat. No. RR037B).

From the resulting cDNA, 5 µg were used for qPCR quantification in a 7500 Fast Real-Time PCR System (Applied Biosystems by Life Technologies, Dormstadt, Germany). Thermal cycling was carried out according to the manufacturer’s recommendations, and the relative expression levels were calculated using the comparative ΔΔCt method. Each reaction contained 5 µL of qPCRBIO SyGreen Blue Mix (PCR Biosystems, London, UK.; Cat. No. P820.15-05) and 50 nM of both forward and reverse primers.

The following primers, obtained from Sigma-Aldrich, were used: P2X1 Fw 5′-GAGAGTCGGGCCAGGACTTC-3′, Rv 5′- GCGAATCCCAAACACCTTGA-3′; P2X2 Fw 5′-TCCCTCCCCCACCTAGTCAC-3′, Rv 5′-CACCACCTGCTCAGTCAGAGC-3′; P2X3 Fw 5′-CTGCCTAACCTCACCGACAAG-3′, Rv 5′-AATACCCAGAACGCCACCC-3′; P2X4 Fw 5′-CCCTTTGCCTGCCCAGATAT-3′, Rv 5′-CCGTACGCCTTGGTGAGTGT-3′; P2X5 Fw 5′-GGATGCCAATGTTGAGGTTGA-3′, Rv 5′-TCCTGACGAACCCTCTCCAGT-3′; P2X6 Fw 5′-CCCAGAGCATCCTTCTGTTCC-3′, Rv 5′-GGCACCAGCTCCAGATCTCA-3′; P2X7 Fw 5′-GGGAGGTGGTTCAGTGGGTAA-3′, Rv 5′-GGATGCTGTGATCCCAACAAA-3′; P2Y1 Fw 5′-AACCGTGATGTGACCACTGA-3′, Rv 5′-TTCAACTTGTCCGTTCCACA-3′; P2Y2 Fw 5′- TGCTGGGTCTGCTTTTTGCT-3′, Rv 5′-ATCGGAAGGAGTAATAGAGGGT-3′; P2Y4 Fw 5′-TCGATTTGCAAGCCTTCTCT-3′, Rv 5′- CCATAGGAGACCAGGGTGAT-3′; P2Y6 Fw 5′-TGCTGCTACC CCCAGTTTAC-3′, Rv 5′- TGGCATAGAAGAGGAAGCGT-3′; P2Y12 Fw 5′-CTGTTTTTTGCTGGGCTCATC-3′, Rv 5′-GCGGATCTGGAAGAAAATCCT3-3′; P2Y13 Fw 5′-GGATGCAGGGCTTCAACAA-3′, Rv 5′- GCAGCTGTGTCATCCGAGTGT-3′; P2Y14 Fw 5′-GGTGGGTTTCGCCTCATGT-3′, Rv 5′-CCTCAGGTGACCGGCATCT-3′; GAPDH Fw 5′-CATCACTGCCACCCAGAAGACTG-3′; Rv 5′-ATGCCAGTGAGCTTCCCGTTCAG-3′.

### 2.5. Recording of Ca^2+^ Channel Currents

The membrane currents through voltage-activated Ca^2+^ channels were measured as previously described [24] using the whole-cell configuration of the patch clamp technique [25]. For current recordings, 2 mM Ba^2+^ (instead of 2 mM Ca^2+^) was used as a charge carrier. The cells were clamped at a −80 mV holding potential and were step depolarized to 0 or +10 mV from this holding potential for 50 ms applied at 30 s intervals to minimize current rundown [26].

Data were acquired at a sample frequency of 20 kHz using the PULSE 8.74 software (Heka Elektronik). The linear leakage and capacitive components were subtracted using a P/4 protocol, and series resistance was 80% compensated. The data analysis was performed using the Igor Pro (Wavemetrics, Lake Oswego, OR, USA) and PULSE programs (Heka Elektronik). All electrophysiological experiments were performed at room temperature (22 ± 2 °C).

### 2.6. Immunohistochemistry of Adrenal Glands

The adrenal glands were fixed with 4% PFA and, after 2 days, were cryoprotected in 30% sucrose solution. Then, they were cut with a slicing microtome into 20 µm slices. After the generous washing of the tissues with PBS and the blocking of the tissues with goat serum, the slices were incubated overnight with primary antibodies—antiCav1.3 (1:200; ACC-003; Alomone Labs; Jerusalem, Israel) and antiCav2.1 (1:200; ACC-001; Alomone labs)—and then with secondary antibodies (1:500). Finally, nuclei were stained with DAPI (1:500), and sections were mounted using ProLong Diamond (Thermofisher Scientific, Waltham, MA, USA).

Images were acquired using a confocal microscope (TCS SP5, Leica), and stacks were made with a step size of 3 µm and a 63X immersion oil objective. The expression of both VDCC subtypes was determined using IntDen with the Fiji ImageJ software.

### 2.7. Data Representation and Statistical Analysis

All data are represented as mean ± SEM. Outlier identification was performed using the GraphPad Prism online tool Grubb’s test (GraphPad Prism Software, San Diego, CA, USA). The normality of the data was assessed using a Kolmogorov–Smirnov test. Depending on whether the data followed a normal distribution or not, the differences between the WT and 3xTg groups were evaluated with a t-student or Mann–Whitney U test, respectively, using GraphPad Prism 5. The results were considered significantly different if the *p* value was lower than 0.05.

## 3. Results

### 3.1. The Modulation of I_Ba_ by ATP in Chromaffin Cells Is Greater in 3xTg than in WT Mice

As described in the introduction, the ATP co-released with catecholamines during exocytosis exerts an autocrine-paracrine modulation of VDCC via G protein coupled receptors in CCs [20,21]. Since this modulation is altered in human pathological processes, such as pheochromocytoma [27], we examined whether ATP could differentially modulate I_Ba_ in CCs with aging and AD pathology. For this purpose, we used WT and 3xTg mice at 2 months (2m) and over 12 months age (>12 m).

In these experiments, I_Ba_ currents were elicited by 10 ms depolarizing pulses to 0 or +10 mV from a holding potential of −80 mV, as indicated in the protocol shown at the top of Figure 1A (below). Figure 1A shows a representative current trace obtained in a CC from a >12 m WT mice. The superfusion of the cell with a solution containing 10 µM of ATP decreased the amplitude of the current peak by approximately 30%. In a similar experiment conducted in 3xTg-AD age-paired CCs, the reduction in the current amplitude was greater (about 50%; Figure 1C).

Figure 1B,D show the normalized data for ATP-induced I_Ba_ modulation in a total of 42 WT and 55 3xTg CCs, respectively. ATP (10 µM) produced a current blockade of 28.11 ± 1.01% in the WT (Figure 1B) and 48.46 ± 1.11% in the 3xTg animals (Figure 1D). In both cases, the application of a short depolarizing pre-pulse to +100 mV preceding the test pulse significantly removed the ATP-induced current blockade (Figure 1B,D), as expected due to the G-protein mediated effect of ATP-induced modulation [21].

Figure 1E shows the concentration–response curves for ATP-induced I_Ba_ blockade obtained in the four experimental groups. A similar concentration dependence for ATP blockade was observed in 2 m animals, with I_Ba_ blockades of 30.35 ± 1.01% (WT) and 29.9 ± 1.51% (3xTg), upon superfusion of the cells with 10 μM ATP (Figure 1F). In CCs from 12m animals, the ATP-induced blockade increased from 28.11 ± 1.10% in WT cells to 48.45 ± 1.10% in 3xTg cells (Figure 1F), indicating a stronger modulatory effect in the second group of animals. At this point, we decided to further characterize whether this variation could be related to differences in the purinergic receptors that mediate the modulatory effects of ATP and/or in the population of VDCCs that are modulated by ATP.

### 3.2. Altered Relative Expression of Different Purinergic Receptors in Adrenal Medulla

The observed differences in the modulation of I_Ba_ from 12 m 3xTg mice compared to their age-paired controls might be due to differences in the relative expression of purinergic receptor subtypes. Therefore, we decided to further explore this possibility by performing RT-qPCR experiments with the entire adrenal medulla from WT and 3xTg animals. In these experiments, the relative expression of the main metabotropic (P2Y) and ionotropic (P2X) receptors was quantified.

As shown in Figure 2A, we found a significant increase in the relative expression of P2Y13R (about three-fold) compared to expression found in control WT animals (1.96 ± 0.2 AU vs. 0.61 ± 0.11 AU). Similarly, the relative levels of P2Y4 receptor mRNA were also significantly increased in 3xTg mice (1.35 ± 0.16 AU) compared to 12m WT mice (0.90 ± 0.09 AU).

When measuring the relative expression of the main ionotropic (P2X) purinergic receptors, we found a significant increase in the mRNA levels of P2X4R (6.51 ± 1.18-fold) and P2X7R (4.14 ± 0.83-fold) in 3xTg mice compared to the age-paired control group. No significant differences were found between the WT and 3xTg-AD groups at 2 m of age (Data not shown)

### 3.3. Increase in P/Q Currents and Decrease in L Currents in 3xTg Mice

The expression of different high-threshold VDCC channel subtypes, including Cav1 (L type) and Cav2 (P/Q, N, and R type), in the membrane of mouse CCs has been reported previously in different studies [20,28,29,30,31]. In fact, L-type channels account for 40% of the total calcium current (I_Ca_), and non-L-type (60%) channels consist of about 25% N type and 35% P/Q type [31].

We used supramaximal concentrations of nifedipine (NIFE, 10 µM), ω-conotoxin-GVIA (GVIA; 1 µM), and ω-agatoxin IVA (AGAIVA; 1 µM), which selectively block the L, N, and P/Q components of inward currents through the VDCCs of CCs [20] to characterize possible differences in their expression related to the progression of AD. To this aim, we initially applied a series of 50-ms depolarizing pulses to +10 mV until the current peak was stabilized. Then, we added the different blockers (i.e., AGAIVA, GVIA, and nifedipine) to quantify the degree of blockade induced by each one. Figure 3A shows a representative I_Ba_ current recording from a WT CC (left panel) and a 3xTg CC (right panel), and the blockade induced by the addition of the different blockers, as indicated. Figure 3B shows the time course of I_Ba_ amplitude and its blockade after the cumulative addition of the different blockers. After blocker washout, only the L component of the I_Ba_ current (NIFE-sensitive) was recovered. These experiments suggest that the L-, N-, and P/Q-types of VDCCs are present in CCs isolated from both WT and 3xTg mice; however, some differences, particularly in the amplitude of the AGAIVA blockade, were observed.

To measure potential differences in the activity of VDCC subtypes, we applied the different drugs separately. Figure 3C shows the averaged pooled results of 22 WT and 23 3xTg CCs, derived from a total number of five animals from each experimental group. In these experiments, GVIA exerted a similar blockade—nearly 25% of the whole-cell current. However, large differences were found in the L and P/Q components. The nifedipine blockade was halved in 3xTg cells compared to WT control cells (from 47.96 ± 2.12% to 21.56 ± 1.38%). In the case of AGAIVA, the I_Ba_ blockade increased from 23.90 ± 1.47% in WT cells to 48.18 ± 1.23 in 3xTg cells, suggesting that a reduction in the L-type current is compensated by a larger I_Ba_ fraction of P/Q channels in 3xTg mice.

### 3.4. In 3xTg Adrenal Medulla There Is an Increase in P/Q Channels and a Decrease in L Subtype Channel Expression

In order to assess whether the differences found in the currents flowing through the different VDCC subtypes in >12 m 3xTg mice correlated with changes in their expression, we measured the mRNA levels of the channels in both WT and 3xTg adrenal medullae using RT-qPCR.

Figure 4A shows the averaged data of the normalized mRNA levels for the different VDCCs found in 3xTg relative to WT. We found that the relative expression of the L subtype of VDCCs was significantly decreased in >12 m 3xTg mice compared to the age-matched control group (0.181 ± 0.04 AU in 3xTg, 1.000 ± 0.188 AU in WT). The opposite was observed for the P/Q subtype, which showed an almost three-fold increase in mRNA levels in 3xTg (2.481 ± 0.396) relative to WT mice (0.886 ± 0.147; Figure 4C).

To correlate the VDCC mRNA levels with their expression in the cell membrane, immunohistochemistry experiments were performed on adrenal gland slices. By using specific antibodies targeting the L and P/Q subtypes of VDCCs, we found a decreased expression of L-type VDCCs in the 3xTg adrenal medulla and an increased expression of the P/Q type. Figure 4B shows representative adrenal slices incubated with L-type (top) and P/Q (bottom) antibodies. Figure 4C displays the average data from the *IntDen* analysis of L-type (top) and P/Q-type (bottom) expression from a total number of five animals of each experimental group. Thus, it seems that the observed alterations in VDCC activity in 3xTg mice may be due to a differential expression of these channels in the CC membrane.

### 3.5. Alterations Observed in mRNA Levels of Purinergic Receptors and Calcium Channels Are Found Also in the Hippocampus

Similar to what occurs in adrenomedullary CCs, hippocampal neurons and microglial cells also express different subtypes of purinergic receptors [32] and VDCCs [33]. Therefore, we questioned whether the alterations we found regarding the mRNA levels of purinergic receptors and VDCCs in the CCs might also be present in the hippocampus.

Similar to our findings from the adrenal medulla, the levels of P2Y13R were also elevated in the hippocampus. However, the same did not occur in case of P2Y4R levels (Figure 5A). Regarding the ionotropic P2X receptors, the observed changes in mRNA levels in the adrenal medulla were similar to those found in the hippocampus, with a higher mRNA content of P2X4R and P2X7R in >12 m 3xTg mice (Figure 5B).

Finally, in the case of VDCC subtypes, the findings from the hippocampus mimicked those found in the adrenal medulla, with higher L-subtype and lower P/Q-subtype mRNA content in > 12 m 3xTg mice compared to the age-matched WT mice (Figure 5C). Therefore, these results suggest that CCs remain a good model to study central nervous system alterations, and that AD is a multifactorial disease that not only affects the CNS, but also the PNS.

## 4. Discussion

The central finding of this investigation was the altered modulation of inward currents through VDCCs that was observed in isolated CCs from the adrenal medulla of >12 m 3xTg mice. The stronger modulation of inward currents through VDCCs seems to be related to the progression of the disease, as it did not appear with normal ageing in >12 m WT mice (Figure 1).

Previous studies from our group have demonstrated that ATP, along with other neurotransmitters and modulators that are co-stored in synaptic vesicles and co-released with catecholamines during the exocytotic process, exert an autocrine-paracrine voltage-dependent modulation of VDCCs via G protein coupled transmembrane receptors in isolated CCs [20,21,22,34] as well as in slices of the adrenal medulla [35,36]. This modulation is manifested by a delay in channel openings (slowing of the activation phase) at low potentials, and can be reversed by the application of strong (+100 mV) depolarizing voltage steps or by the application of brief depolarizing pulses repeated at high frequency (150), as happens in the high-frequency action potential (AP) trains that occur during stress-mimicking conditions. From a physiological point of view, it seems that the autocrine inhibition of the VDCCs acts primarily as negative feedback to regulate the amount of secretion at resting conditions. With increased firing rates, the autocrine inhibition may still act to prevent excessive catecholamine release; however, when under intense AP stimulation (i.e., in response to a stressful situation), a large surge of catecholamines is required. CCs are able to overcome this inhibition by facilitating VDCC currents in an activity-dependent manner [19,20,28].

It has been also described that this autocrine-paracrine modulation of VDCCs can be altered in certain pathological human processes. This is the case for pheochromocytoma, a tumor arising from CCs in the adrenal medulla and sympathetic paraganglia that can synthesize and secrete catecholamines. While the most common sign of pheochromocytoma is hypertension, additional symptoms include episodes of headaches, palpitation, anxiety, and sweating related to excess catecholamine secretion [37]. One of the few physiological studies that has been conducted on cultured human pheochromocytoma cells [27] revealed that they express densities and proportions of L, N, and P/Q types of voltage-gated Ca^2+^ channels similar to those of normal human CCs [38]. Nonetheless, the extent of the downmodulation of Ca^2+^ currents by ATP and opioids is quite heterogeneous in these cells. It was found that normal CCs coexist with pathological CCs that escape the autocrine-paracrine downmodulation of Ca^2+^ channels and, hence, may produce abnormal Ca^2+^ signals and catecholamine hypersecretion.

Alterations in the autocrine-paracrine modulation of catecholamine release have been also described in hypertension. A study conducted with spontaneously hypertensive rats (SHRs) showed a greater modulation of I_Ba_ by ATP in SHR compared with normotensive rat CCs [24]. In addition, it was also found that SHR CCs had a lower density of L-type VDCCs that was compensated for by a higher density of PQ-type channels, an effect similar to that found in the present study in CCs isolated from aged 3xTg mice (see Figure 3C).

The expected increase in catecholamine release under stressful situations in AD patients as a consequence of the reversion of VDCC modulation under these conditions could explain the appearance of non-cognitive symptoms as AD progresses. It has been determined that in advanced AD stages, in addition to the initial cognitive symptoms, non-cognitive behavioral problems, particularly agitation and aggression, become prevalent in AD patients [39]. As the noradrenergic system has been implicated in the disorder of behavior in animals, as well as other psychiatric disorders, it has been suggested that norepinephrine may likely play a role in the non-cognitive behavioral disturbances associated with AD [40]. As a result of this noradrenergic overactivation, AD patients would no longer be able to focus their attention, and their mechanisms of coping with stressful stimuli would be compromised. Even in the absence of stressful stimuli, the noradrenergic system would be active. This may account for the increased aggression displayed in many AD patients [39].

To further identify the mechanisms involved in the increased modulation of catecholamine secretion observed in 3xTg CCs that occurs with AD progression, we decided to study possible changes in the two main elements involved in ATP-induced modulation, i.e., the purinergic receptors and the VDCCs.

Firstly, regarding the expression of purinergic receptors, our data indicate that upon the progression of AD in 3xTg mice, a significant three-fold increase in the relative expression of P2Y13 receptor mRNA compared to expression in control WT animals occurs (Figure 2). The modulation of catecholamine release by ATP is thought to be mediated by a P2Y purinergic receptor [21]. Unfortunately, the absence of selective pharmacology (i.e., selective agonists or antagonists) precludes the characterization of the P2Y receptor subtypes implicated in this modulation. It should be noted that the levels of P2Y13 receptor mRNA were also increased in hippocampus (Figure 5). A cytoprotective activity mediated through an increase in the activity of heme oxygenase and by the enhanced expression of transcription factor Nrf2, which prevents neuronal death due to oxidative stress, has been described for P2Y13 receptors [41,42]; therefore, the observed increased expression of this receptor subtype could be interpreted as a compensatory, neuroprotective mechanism that occurs during AD progression [43].

In parallel to the determination of P2Y receptor mRNA levels, although purinergic ionotropic (P2X) receptors do not appear to be involved in the modulating effects of ATP, we decided to also characterize possible changes in these receptors that could be related to the progression of AD. In these experiments, we found a significant increase in mRNA levels of P2X4 and P2X7 receptors in 12-month-old 3xTg mice with respect to the age-matched control animals, both in the adrenal medulla tissue (Figure 2) and hippocampus (Figure 5). P2X4 and P2X7 receptors seem to be involved in inflammatory signals in neurodegenerative diseases. Among other functions, it has been reported that microglial activation results in the upregulation of P2X4 and P2X7 [44]. 

Other authors have reported an increase in these receptors in different neurodegenerative diseases, such as Parkinson’s disease [45]; they have therefore been proposed as potential therapeutic targets for the treatment of these diseases. Regarding the P2X7 receptor, there are studies that have reported its relationship with the formation of Aβ plaques in AD patients, in which this receptor has been found to be overexpressed [46,47]. The P2X7 receptor, as previously discussed, has an important role in neuroinflammation; thus, it may be implicated in the maintenance of the characteristic pro-inflammatory environment in AD pathophysiology by releasing pro-inflammatory interleukins and cytokines via caspase 1 activation [48].

Finally, regarding possible modifications in the expression of VDCCs related to AD progression in 3xTg mice, we found a significant decrease in both the inward currents through VDCCS and the expression levels of L-type VDCCs, which was compensated by an increase in P/Q-type VDCCs (Figure 3C and Figure 5C). It should be noted that the voltage-dependent modulation induced by ATP and other neurotransmitters affects non-L-type (mainly P/Q) VDCCs [20,27,33]. Moreover, as previously mentioned, similar changes in the relative expression of VDCC subtypes have been reported in other pathologies, such as hypertension, where an altered regulation of the sympathoadrenal axis is also observed [38].

Despite the contributions that this work provides to the field, there are three major limitations that should be addressed in future research. First, although the 3xTg mouse model fulfills many pathophysiological traits present in AD patients, there may be species-related differences that prevent the application of these observations to human pathology. Therefore, further investigations should be performed using human samples from AD patients’ necropsy, or in models including induced pluripotent stem cells derived from AD patients.

On the other hand, in order to address if the observed alterations of the purinergic receptors’ expression at the transcriptional level lead to differences at the translational level, and also to see their location in the cell membrane, it would be desirable to conduct some immunohistochemistry or Western blot experiments. Finally, further research into the synthesis of pharmacological compounds that provides selective agonists and antagonist for the different P2Y receptors would be very helpful for elucidating the specific contribution of the different metabotropic purinergic receptors to the modulation of VDCCs.

## 5. Conclusions

In conclusion, an impairment in the sympathoadrenal axis seems to occur with the progression of AD in the 3xTg mouse model of this disease. This alteration is evidenced by the increased autocrine-paracrine modulation of VDCC activity, which is reversed during stressful conflicts and may produce an increase in catecholamine secretion that, in turn, could be involved in the appearance of non-cognitive symptoms as AD progresses.

## Figures and Tables

**Figure 1 biology-11-00511-f001:**
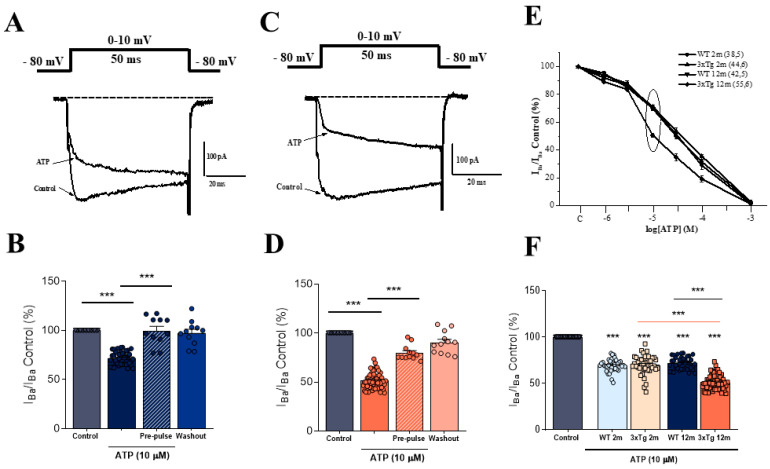
Modulation of I_Ba_ by ATP in wild-type (WT) and 3xTg chromaffin cells. Whole-cell I_Ba_ currents were elicited by 50 ms depolarizing test pulses to 0–+10 mV applied at 30 s intervals from a holding potential of −80 mV. (**A**,**C**) Representative I_Ba_ traces obtained in the absence (control) and the presence of 10 μM ATP obtained from a WT (**A**) and 3xTg (**C**) CC. (**B**,**D**) Averaged pooled results obtained in these series of experiments in the 12 m WT group. Each dot corresponds to a single WT (**B**) and 3xTg (**D**) cell. Data (ordinate) were normalized as percent of control I_Ba_ (100% in the absence of ATP) within each individual cell. Bars represent mean ± SEM. (**E**) Concentration–response curves for the ATP-induced blockade of I_Ba_ obtained in the different experimental groups. (**F**) Averaged data on I_Ba_ blockade induced by 10 μM ATP in each experimental group (each dot corresponds to a single CC). A one-way ANOVA statistical analysis with multiple comparisons was conducted to evaluate differences among all experimental groups. *** *p* < 0.001.

**Figure 2 biology-11-00511-f002:**
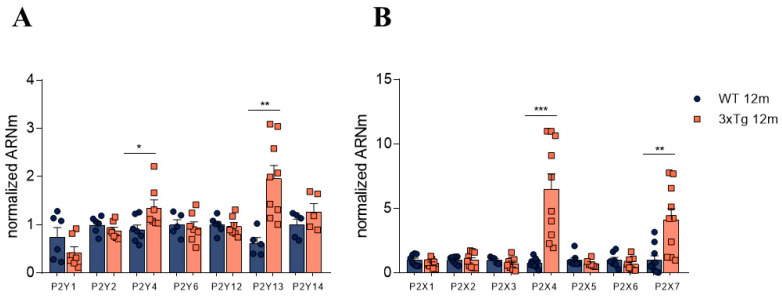
Normalized mRNA levels of the indicated receptors in adrenal medulla tissue obtained from WT and 3xTg mice that were more than 12 months old. (**A**) Normalized mRNA levels of the main metabotropic purinergic receptors (P2Y receptors). (**B**) Normalized levels of ionotropic purinergic receptors (P2X receptors). Each dot corresponds to the result of both mouse medullae and the bars represent mean ± SEM. A Student’s *t*-test was used to statistically compare the WT and 3xTg groups. * *p* < 0.05; ** *p* < 0.01; *** *p* < 0.001.

**Figure 3 biology-11-00511-f003:**
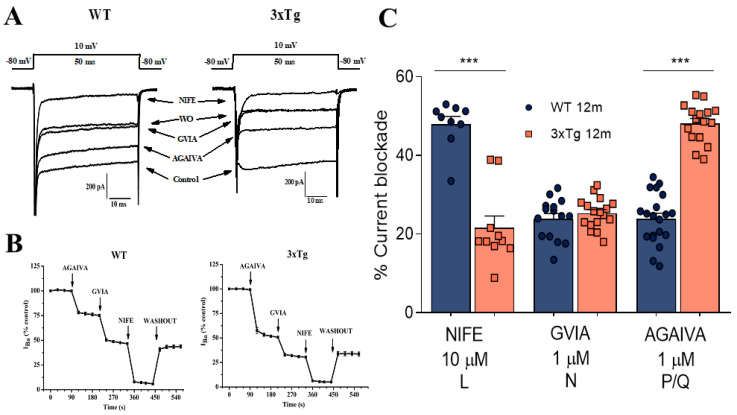
Pharmacological dissection of I_Ba_ subcomponents linked to different VDCC subtypes in chromaffin cells from WT and 3xTg mice older than 12 m. Cells were voltage-clamped at −80 mV and the I_Ba_ current was generated by 50-ms depolarizing pulses to +10 mV applied at 30 s intervals. The cells were sequentially fast-perfused with an extracellular solution containing 2 mM Ba^2+^ in the absence (control current) and the presence of 1 μM ω-agatoxin IVA (AGAIVA), 1 μM ω-conotoxin GVIA (GVIA), and 10 μM nifedipine (NIFE); in these representative experiments, the blockers were cumulatively added (panel **B**). (**A**) Representative I_Ba_ traces from a WT cell (left) and a 3xTg cell (right); each trace was selected from the time course of I_Ba_ traces, after 2 min of cell perfusion with each blocker and once the steady-state equilibrium of the new level of the peak I_Ba_ current was reached. (**C**) Pooled averaged results on the extent of peak I_Ba_ blockade (at equilibrium) elicited by each blocker (abscissa); note that only one blocker was tested in each cell. The percent blockade was calculated by normalizing the control I_Ba_ to 100% and expressing the I_Ba_ blockade elicited by each compound within each individual cell. Bars represent mean ± SEM. Student’s t-test was used to statistically compare the WT and 3xTg groups. *** *p* < 0.001.

**Figure 4 biology-11-00511-f004:**
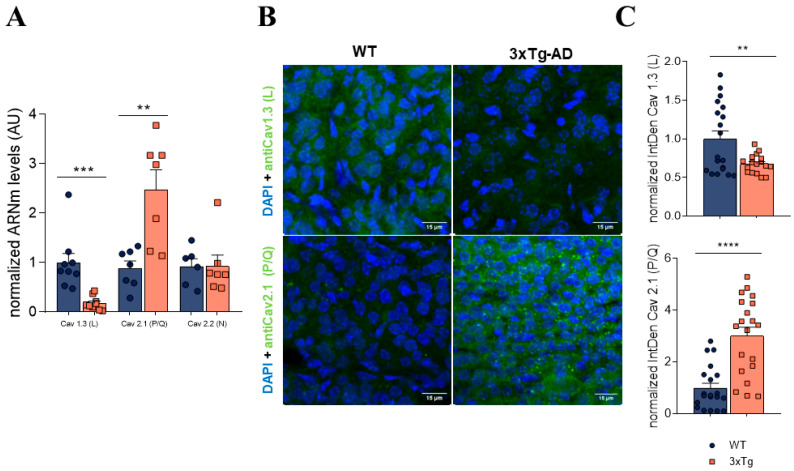
Analysis of mRNA levels and the expression of different VDCC subtypes in adrenal medullae of WT and 3xTg mice over 12 m age. (**A**) Normalized mRNA levels of different VDCCs in adrenal medullae from WT and 3xTg mice > 12 m. Both medullas of the same mouse were homogenized, so each dot represents an individual mouse. (**B**) Representative image of WT and 3xTg adrenal medulla stained with anti-Cav1.3 (top) and anti Cav2.1 (bottom) + DAPI (nuclei). (**C**) IntDen quantification of calcium channel expression. Each dot represents an image analyzed, and the bars represent mean ± SEM. A Student’s *t*-test was used to statistically compare the WT and 3xTg groups. ** *p* < 0.01; *** *p* < 0.001; **** *p* < 0.0001.

**Figure 5 biology-11-00511-f005:**
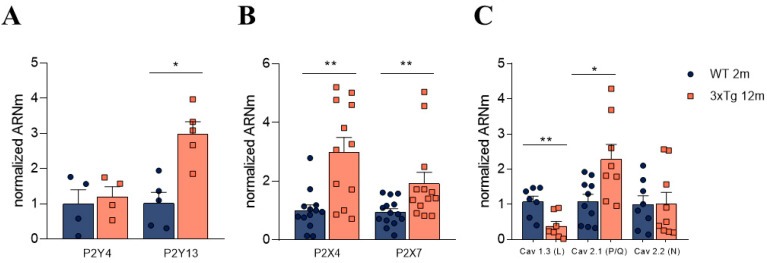
Analysis of mRNA levels of different purinergic receptors and calcium channels in the hippocampus of WT and 3xTg mice over 12 m of age. Graphs show normalized mRNA levels in the hippocampus from >12 m WT and 3xTg mice. (**A**) Normalized mRNA levels of metabotropic (P2Y) purinergic receptors. (**B**) Normalized mRNA levels of ionotropic (P2X) purinergic receptors. (**C**) Normalized mRNA levels of calcium channel subtypes. Each dot represents a single hippocampus isolated from one mouse, and the bars represent mean ± SEM. A Student’s *t*-test was used to statistically compare the WT and 3xTg groups. * *p* < 0.05; ** *p* < 0.01.

## Data Availability

The data presented in this study are available on request from the corresponding author. The data are not publicly available due to privacy.

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
