# Peer review of "Alterations of the Sympathoadrenal Axis Related to the Development of Alzheimer’s Disease in the 3xTg Mouse Model"

_biology, 2022, doi:10.3390/biology11040511_

Round 1

Reviewer 1 Report

In the present study, the authors explore the involvement of adrenal chromaffin cells in the development of Alzheimer's Disease in the 3xTg mouse model. The study begins from an observation of the effects of extracellular ATP on inward currents mediated by VDCCs voltage-clamped CCs, from which it was determined that ATP has a stronger modulatory effect on CCs from 3xTg mice. Differences between 3xTg and WT CCs are then further explored: purinergic receptors P2Y13R and P2Y4 are signficantly upregulated in 3xTg CCs, as are P2X4R and P2X7R. The authors also investigated which VDCCs were most involved in this sensitization process, and observed that changes occurring in adrenal 3xTg CCs reflected pretty well the pathological occurrences in the hippocampus. This is a very elegant demonstration of the involvement of the peripheral nervous system in AD pathology.

The methodologies applied by the authors are appropriate, as is the statistical analysis of obtained data. The conclusions follow smoothly from the obtained results. The manuscript is quite well-written, however a wide-ranging english spellcheck is recommended, as there are quite a few typos and awkward turns of phrase, such as in page 5: "superfusion of the cell with a solution containing 10μM of ATP blocked slowed-down and decreased about 30% de peak current". This and a few other passages could be rephrased in order to improve manuscript readability.

All in all, I recommend this article for publication in Biology, provided such minor improvements to the text take place.

Author Response

We appreciate the useful comments and suggestions from the reviewer that undoubtedly helps to improve our manuscript.

We have improved the english spelling to the best of our knowledge, correcting some typographical mistakes and complex sentences (including the sentence on ATP effects, as indicated).

We hope that this new version of the manuscript will seem more readable.

Reviewer 2 Report

  1. This manuscript is well organized but still needs a proofreading in English grammar and some spellings.
  2. The format of the manuscript can be improved for better understanding. 
  3.  The major defect of the design is that the author should discuss more on the microglial purinergic receptors. 

Author Response

We appreciate the useful comments and suggestions from the reviewer that undoubtedly helps to improve our manuscript.

We have improved the english spelling to the best of our knowledge, correcting some typographical mistakes and complex sentences. So, we hope that this new version of the manuscript will be more easily understandable.

We have included a new paragraph in the Discussion section to comment about the role of P2X7 receptor at microglial level (highlighted in yellow in the new version of the manuscript), as suggested.

Reviewer 3 Report

THe manuscript is well written. Authors described results very well,  and the context in introduction as well as hypothesis are clear. 

The discussion is also clear and authors made clear link with the literature.

I only have two suggestions:

1-  Figure 4B I would suggest to take higher quality image since they are blurred maybe the because of the format. 

2- I would suggest to discuss some limitations of the study in the discussion section 

Author Response

We appreciate the useful comments and suggestions from the reviewer that undoubtedly helps to improve our manuscript.

We have improved the quality of Fig. 4B, as indicated by the reviewer.

We have included a new paragraph in the Discussion section to comment about some possible limitations of our study (highlighted in yellow in the new version of the manuscript), as suggested.